# Oral Administration of a Select Mixture of *Lactobacillus* and *Bacillus* Alleviates Inflammation and Maintains Mucosal Barrier Integrity in the Ileum of Pigs Challenged with *Salmonella* Infantis

**DOI:** 10.3390/microorganisms7050135

**Published:** 2019-05-15

**Authors:** Xiao Liu, Bing Xia, Ting He, Dan Li, Jin-Hui Su, Liang Guo, Jiu-feng Wang, Yao-Hong Zhu

**Affiliations:** College of Veterinary Medicine, China Agricultural University, Beijing 100193, China; jizi93112@163.com (X.L.); xiabing26@163.com (B.X.); ting_he413@163.com (T.H.); emma6861@163.com (D.L.); sujh0209@gmail.com (J.-H.S.); binghaitiankong20@163.com (L.G.); jiufeng_wang@hotmail.com (J.-f.W.)

**Keywords:** *Lactobacillus*, *Bacillus*, *Salmonella* Infantis, intestinal mucosal barrier, pig

## Abstract

*Salmonella* is important as both a cause of clinical disease in swine and as a source of food-borne transmission of disease to humans. *Lactobacillus* and *Bacillus* are often used as antibiotic substitutes to prevent *Salmonella* infection. In this study, we evaluated the effects of a select mixture of *Lactobacillus johnsonii* L531, *Bacillus licheniformis* BL1721 and *Bacillus subtilis* BS1715 (LBB-mix) in prevention of *Salmonella*
*enterica* serovar Infantis infection in a pig model. LBB-mix was orally administered to newly weaned piglets for seven days before *S*. Infantis challenge. LBB-mix pretreatment ameliorated *S*. Infantis-induced fever, leukocytosis, growth performance loss, and ileal inflammation. Pre-administration of LBB-mix reduced the number of *Salmonella* in the feces but increased the number of goblet cells in the ileum. *S*. Infantis infection resulted in an increase in cell death in the ileum, this increase was attenuated by LBB-mix consumption. Claudin 1 and cleaved caspase-1 expression was decreased in the ileum of pigs challenged with *S*. Infantis, but not in pigs pretreated with LBB-mix. In conclusion, our data indicate that a select LBB-mix has positive effects on controlling *S*. Infantis infection via alleviating inflammation and maintaining the intestinal mucosal barrier integrity in pigs.

## 1. Introduction

*Salmonella* is a common source of food or water-borne infection and causes a wide range of clinical disease in humans and animals. In 2017, a total of 91,662 confirmed human salmonellosis cases were reported in the European Union, and the top five most commonly reported serovars were *Salmonella enterica* Enteritidis, *S*. Typhimurium, monophasic *S*. Typhimurium, *S*. Infantis and *S*. Newport [1]. *S*. Infantis causes acute foodborne gastroenteritis, and *S*. Infantis-contaminated pork products are frequent causes of human salmonellosis [2]. The incidence of *S*. Infantis in pigs has increased in recent years [3]. In China, the prevalence of *S*. Infantis in pigs with *Salmonella*-associated diarrhea was 3.85% from 2014 to 2016 [4]. Conventional salmonellosis control strategies have relied on antibiotics. However, due to the emergence of drug-resistant strains, many countries have already restricted antibiotic use in animal agriculture. In 2006, the European Union banned the use of antimicrobial growth promoters in animal food and water [5]. In recent years, different intervention strategies such as vaccination, antimicrobial peptides, nutritional supplements, bacteriophages, and probiotics have been used to control *Salmonella* infection as antibiotic alternatives [6].

Different probiotic strains have different effects to prevent or treat diseases. Combined use of probiotic strains has been suggested to have greater efficacy than single strains because they may integrate the effects of different individual strains [7]. *Lactobacillus* and *Bacillus* are the most commonly used probiotic agent to improve growth performance, gut health and regulate the immune system in pigs. Our previous study showed that *Lactobacillus johnsonii* L531 reduced pathogen load and helps maintain short-chain fatty acid levels in the intestines of pigs challenged with *S*. Infantis [8]. Oral administration of a select mixture of *Bacillus* probiotics ameliorated *Escherichia coli*-induced enteritis through preventing loss of intestinal epithelial barrier integrity in weaned pigs [9]. There was evidence that oral supplementation with *Bacillus* strains increased the abundance of *Lactobacillus* species in pigs [10,11]. Research on the combined use of *Lactobacillus* and *Bacillus* to prevent *Salmonella* infection are still rare.

The intestinal mucosal barrier is the first line of host defense against enteric pathogens, which mainly includes the outer mucus layer, the central single cell layer with specialized epithelial cells, and the inner lamina propria where innate and adaptive immune cells reside. Goblet cells (GCs), known for their apical release of mucin 2 (Muc2), are the predominant secretory epithelial cells lining the luminal surface of the mammalian gastrointestinal tract. Previous studies have shown that oral administration of *Lactobacillus acidophilus* obviously alleviated *S*. Typhimurium-induced goblet cells loss in mice [12]. A select mixture of *Bacillus licheniformis* and *Bacillus subtilis* ameliorated *E. coli*-induced decreases in the number of ileal GCs via increasing Atoh1 mRNA expression in newly weaned pigs [10]. Tight junctions (TJs), as a barrier against solute diffusion through the intercellular space, consist mainly of transmembrane protein complexes (e.g., claudin 1 and occludin) and the cytosolic proteins zonula occludins (e.g., ZO-1) [13]. As a crucial component of the epithelial barrier, they are often targeted by pathogenic bacteria, aiding infection and the development of disease [14]. *L. acidophilus* attenuated *Salmonella*-induced stress of epithelial cells by modulating TJs genes [15]. *B. subtilis* significantly enhanced intestinal barrier integrity by up-regulating tight junction protein expression [16].

*Salmonella* is recognized by the innate immune system and drives pro-inflammatory responses through activation of the inflammatory caspase (caspase-1, -4, -5 in humans; and caspase-1, -11 in mice). Caspase-1, the most fully characterized inflammatory caspase, cleaves the inactive cytokine precursors pro-interleukin-18 (IL-18) and pro-IL-1β, to yield their active secreted forms [17,18]. Caspase-4 is activated when recognizing intracellular lipopolysaccharide [19]. Activated inflammatory caspases also cleave pro-gasdermin-D (GSDMD) and the gasdermin-N domain of GSDMD subsequently forms membrane pores to drive pyroptosis, a lytic form of cell death that triggers inflammation [20,21,22]. Our previous study demonstrated that pre-incubation with *Lactobacillus rhamnosus* GR-1 decreased *E. coli*-induced bovine mammary epithelial cell pyroptosis [23]. However, the role of probiotics in controlling of *Salmonella*-induced intestinal cell pyroptosis is unclear.

The aim of this study was to determine the ability of a select mixture of *Lactobacillus* and *Bacillus* to prevent *S*. Infantis infection and to explore its possible mechanism. We investigated the effects of a select mixture of the potential probiotics *Lactobacillus johnsonii* L531, *Bacillus licheniformis* BL1721 and *Bacillus subtilis* BS1715 (LBB-mix) on *S*. Infantis infection in newly weaned piglets. We hypothesized that oral administration of LBB-mix alleviates clinical symptoms and inflammation by maintaining the intestinal mucosal barrier integrity and reducing intestinal cell death during *S*. Infantis infection.

## 2. Materials and Methods

### 2.1. Bacterial Strains

A total of 27 *Bacillus* strains were isolated from the feces collected from 18 healthy newly weaned piglets. Based on the results of in vitro experiments, *Bacillus licheniformis* BL1721 and *Bacillus subtilis* BS1715 were selected as putative probiotic strains. Previous studies have demonstrated the probiotic properties of *L. johnsonii* L531 [8].

*S*. Infantis CAU1508 was isolated from the intestinal contents of weaned pigs with diarrhea, as previously described [24].

### 2.2. Characterization of B. licheniformis BL1721 and B. subtilis BS1715 In Vitro

Characteristics of the two *Bacillus* isolates were evaluated in comparison to *B. licheniformis* DSM 5749 and *B. subtilis* DSM 5750, two strains isolated from an EU-authorized animal probiotic mixture Bioplus^®^ YC (Hørsholm, Denmark).

Tolerance to low pH and pig bile salts were evaluated by inoculating 10^8^ CFU mid-log phase *Bacillus* into Luria-Bertani (LB) broth at pH 3.0 or 0.3% bile salts, respectively. The number of residual viable bacteria was determined at various time points (0 h, 1 h, 2 h and 4 h). Bacteria were diluted appropriately and plated onto LB agar. Plates were incubated at 37 °C for 24 h, after which the number of CFU/mL was calculated.

Bacterial adhesion capacity was assessed using porcine jejunal epithelial J2 cells (IPEC-J2, ACC701, Deutsche Sammlung von Mikroorganismen und Zellkulturen). The IPEC-J2 cells (5 × 10^5^ cells/well) were seeded onto a 6-well trans-well collagen-coated polytetrafluoroethylene (PTFE) filter (0.4 µm pore size, 4.7 cm^2^ growth area, Corning Costar Corp., Cambridge, MA). Confluent cell monolayers were treated with *Bacillus* strains (5 × 10^7^ CFU) for 3 h, 6 h, 9 h, then washed three times with PBS to remove non-adherent bacteria and treated with 0.05% trypsin for 10 min at 37 °C. Cells were harvested by centrifugation for 10 min at 4000× *g* and lysed with 100 µL of 0.2% Triton X-100 in sterile water. The populations of *Bacillus* were determined on LB agar plates.

Trypan blue staining was used to investigate the cell death. The IPEC-J2 cells (5 × 10^5^ cells/well) were seeded onto a 6-well trans-well collagen-coated PTFE filter. Confluent cell monolayers were treated under one of six conditions, as follows: (i) Dulbecco’s Modified Eagle’s Medium, (Gibco, Grand Island, NY), (ii) *B. licheniformis* BL1721 (5 × 10^7^ CFU), (iii) *B. subtilis* BS1715 (5 × 10^7^ CFU), (iv) *S*. Infantis (5 × 10^6^ CFU), (v) preincubation with *B. licheniformis* BL1721 (5 × 10^7^ CFU) and exposed to *S*. Infantis (5 × 10^6^ CFU), or (vi) preincubation with *B. subtilis* BS1715 (5 × 10^7^ CFU) and exposed to *S*. Infantis (5 × 10^6^ CFU). Cells were harvested by digestion with 0.25% trypsin and centrifugation for 10 min at 4000× *g*. The harvested cells were mixed with 0.4% Trypan blue dye (Invitrogen, Carlsbad, CA). Counted the blue cells under a microscope. Results are presented as the ratio of the number of dead cells to the total number of cells. Data are presented as the mean ± standard error of the mean (SEM) of three independent experiments.

### 2.3. Ethics Statement

All experimental animals were treated in accordance with the Guidelines for Laboratory Animal Use and Care from the Chinese Center for Disease Control and Prevention and Rules for Medical Laboratory Animals from the Chinese Ministry of Health, under protocol AW09059102-2, approved by the Animal Ethics Committee of China Agricultural University (approval date: 17 November 2017). All animals used in this study were housed at the experimental facility of the College of Veterinary Medicine, China Agricultural University.

### 2.4. Animals

A total of 24 (Landrace × Large White) piglets were used in this study. These piglets weaned at 21 days of age and weighed 4.94 ± 0.18 kg. Prior to the trial, no clinical signs of diarrhea or other diseases were observed in any of the piglets. Feed and water were provided ad libitum.

### 2.5. Animal Experiments

On the day of weaning (day 0), the pigs were assigned to three groups (*n* = 8 per group) based on weight. Prior to initiation of the study, all piglets were determined to be free of *Salmonella* by analysis of feces. Each group received a different treatment, as follows: (i) Control (CN) group, oral administration of 10 mL sterile physiological saline for 8 days; (ii) *S*. Infantis (SI) group, oral administration of 10 mL of sterile physiological saline for 7 days and oral challenge with *S*. Infantis on day 8 (10^11^ CFU/mL, 10 mL); (iii) potential probiotic mixture + *S*. Infantis (PS) group, oral administration of 10 mL potential probiotic mixture (*L. johnsonii* L531, 10^8^ CFU/mL; *B. licheniformis* BL1721, 4 × 10^5^ CFU/mL and *B. subtilis* BS1715, 4 × 10^5^ CFU/mL) for 7 days and oral challenge with *S*. Infantis on day 8 (10^11^ CFU/mL, 10 mL). Pigs of each group were housed separately in pens. Feed intake, body weight (BW), clinical symptoms and diarrhea score were recorded every day. On day 13 (5 days post-infection), pigs in these three groups were sacrificed and tissue samples were immediately collected.

### 2.6. Clinical Examinations and Microbiological Analyses

Rectal temperature was measured twice daily, at 7:30 A.M. and 7:30 P.M. respectively. Fecal samples were collected on days 1, 4, 8 (0, 6, and 12 h), 9, 10, 11 and 12 after weaning for each group. The severity of diarrhea was also scored as previously described [24]. One gram of fresh fecal sample was homogenized in 9 mL of sterile saline solution, and suitable dilutions of the homogenates were then plated onto selective medium. The groups of fecal microbes studied and the selective mediums (Beijing Land Bridge Technology Co., Beijing, China) employed were as follows: (i) *Salmonella*, Xylose lysine tergito4 (XLT4) agar; (ii) *Lactobacilli*, deMan, Rogosa, Sharpe (MRS) agar; (iii) *Bacillus* spores, Luria-Bertani (LB) agar (inoculated after heat treatment at 80 °C for 15 min); (iv) bifidobacteria, Tryptone Phytone Yeast (TPY) agar; (v) coliforms, eosin–methylene blue (EMB) agar; (vi) enterococci, Pfizer agar. The plates for XLT4 agar, LB agar, EMB agar, and Pfizer agar were incubated for 24 h at 37 °C under aerobic conditions, whereas MRS agar, TPY agar plates were incubated under anaerobic conditions for 48 h at 37 °C. Results were presented as log_10_ CFU/g feces, and all counts were performed in triplicate.

### 2.7. Differential Blood Leukocyte Count

The total white blood cell count was determined using a semiautomated blood cell counter (MEK-6318K; Nihno Kohden, Inc., Tokyo, Japan) and monitored by microscopic differentiation. The proportions of segmented neutrophils, banded neutrophils and lymphocytes were expressed as a percentage of the total number of leukocytes.

### 2.8. Histopathologic Scoring

The proximal, middle, and distal segments (approximately to 10 × 15 × 3 mm) of the jejunum and ileum were rinsed with sterile physiological saline and then fixed in 4% paraformaldehyde. Intestinal pathology was evaluated on hematoxylin and eosin-stained jejunal and ileal sections by a single blinded scorer using a validated scoring system [24]. The summation of the scores for each parameter provided an overall inflammation score for each sample, with a range of 0–15 in the jejunum and 0–18 in the ileum. No less than three separate sections of each sample were examined. Hematoxylin and eosin-stained tissues were visualized and photographed using an Olympus BX41 microscope (Olympus, Tokyo, Japan) equipped with a Canon EOS 550D camera head (Canon, Tokyo, Japan).

### 2.9. Alcian Blue/Periodic Acid–Schiff Staining and Subsequent Quantification

For mucin-containing GC analysis, ileal sections were stained with alcian blue (AB) for acidic mucin at pH 2.5 or periodic acid–Schiff (PAS) stain for neutral mucin detection (Solarbio, Beijing, China), as described by the manufacturer. For quantification of AB/PAS staining, each segment was visualized and photographed using an Olympus BX41 microscope equipped with a Canon EOS 550D camera head. Positively stained GCs and other absorptive epithelial cells were counted in a blind manner in a total of 10 crypts per piglet. Results are presented as the average ratio of GC count to the count of GCs plus other absorptive epithelial cells.

### 2.10. Analysis of DNA Fragmentation

Intestinal tissue samples were evaluated for DNA fragmentation by terminal deoxynucleotidyl transferase dUTP nick end labeling (TUNEL), employing the In Situ Fluorescein TUNEL Cell Apoptosis Detection Kit (Transgen, Beijing, China). Briefly, intestinal samples were permeabilized with 0.1% Triton X-100 for 10 min. Subsequently, tissues were incubated with TUNEL reactant at 37 °C in the dark for 1 h and washed three times. Sections were examined under an Olympus BX41 microscope equipped with a Canon EOS 550D camera head. Fluorescence intensity was quantified with the Image-Pro Plus 6.0 software (Media Cybernetics, Silver Spring, MD).

### 2.11. Western Blotting

Proteins were extracted from the ileum as previously described [24]. The primary antibodies were as follows: Anti-claudin 1 (1:500 dilution, 13050-1-AP, Proteintech Group, Chicago, IL, USA), anti-occludin (1:1000 dilution, ab31721, Abcam, Cambridge, UK), anti-caspase-1 (1:200 dilution, sc-56036, Santa Cruz Biotechnology, Dallas, TX, USA), anti-caspase-4 (1:1000 dilution, GTX113639, GeneTex, San Antonio, TX, USA), anti-GSDMD (1:500 dilution, sc-81868, Santa Cruz Biotechnology) and anti-Glyceraldehyde-3-phosphate dehydrogenase (GAPDH), (1:1000 dilution, 60004-1-Ig, Proteintech Group). Horseradish peroxidase-conjugated secondary antibodies used were goat anti-mouse IgG (1:5000 dilution, SA00001-1, Proteintech Group) or goat anti-rabbit IgG (1:5000 dilution, SA00001-2, Proteintech Group). The bands were visualized using a Tanon-5200 gel image system (Tanon, Shanghai, China). The intensity of bands was quantified by densitometric analysis using ImageJ software (National Institutes of Health, Bethesda, MD, USA).

### 2.12. Statistical Analysis

Statistical analyses were conducted using SPSS 22.0 software (SPSS Inc., Chicago, IL, USA). For comparisons of the mean values between the three groups, the statistical significance of differences was tested using ANOVA procedures, following Tukey’s honestly significant difference post hoc test. Statistical evaluation of the incidence of diarrhea was carried out using Pearson’s chi-squared test. GraphPad Prism 7.0 software (GraphPad Software Inc., San Diego, CA, USA) was used to process the data. The data are expressed as the mean ± SEM. *p* values: * *p* < 0.05; ** *p* < 0.01; *** *p* < 0.001.

## 3. Results

### 3.1. In Vitro Probiotic Characteristics of B. licheniformis BL1721 and B. subtilis BS1715

There was no significant difference in the acid and bile salts tolerance between the *B. licheniformis* BL1721, *B. subtilis* BS1715, and the two strains isolated from the Bioplus^®^ YC (Figure 1A,B). The number of adhered *B. licheniformis* BL1721 was 1.31 × 10^4^ ± 9.50 × 10^2^ CFU (the mean ± SEM) at 3 h after inoculation, and increased to 4.06 × 10^4^ ± 5.46 × 10^3^ CFU at 9 h. The number of *B. subtilis* BS1715 was 1.13 × 10^4^ ± 1.10 × 10^3^ CFU at 3 h after inoculation, and increased to 1.86 × 10^4^ ± 2.28 × 10^3^ CFU at 9 h (Figure 1C). Both isolates and commercial strains have the similar adhesion ability to IPEC-J2 cells. *S*. Infantis significantly caused IPEC-J2 cell death at 6 h after infection, and the proportion of dead cells increased with the time of *Salmonella* infection. Pre-incubation with *Bacillus* isolates significantly reduced cell death at 12 h after *S*. Infantis challenge (*p* < 0.001 and *p* < 0.001, respectively). Cells incubated with *Bacillus* alone had a low percentage of death (Figure 1D).

### 3.2. Clinical Symptoms and Growth Performance

Before *S*. Infantis challenge, all pigs exhibited a normal rectal temperature (39.22 ± 0.28 °C). The rectal temperature in the SI group peaked at 24 h after challenge (40.37 ± 0.35 °C), and it remained higher until 96 h. Although the rectal temperature in the PS group was higher than the CN group at 24 h (40.10 ± 0.42 °C, *p* = 0.001), there were no differences at other times (Figure 2A).

Before *S*. Infantis challenge, three groups of pigs had a lower diarrhea rate. During *S*. Infantis infection, the incidence of diarrhea in SI and PS pigs was increased (Appendix A). At 6 h after *S*. Infantis challenge, all pigs in the SI and PS groups had diarrhea (fecal scores ≥4), and this phenomenon continued five days after infection (Figure 2B). At 12 h after challenge, pigs in the SI group, not the PS group, had a higher diarrhea score compared with the CN group (*p* = 0.016). Although piglets in the SI group had higher scores until 72 h, there was no significant difference between the SI and PS group at other points.

No significant differences were observed in the initial and final body weight between the three groups (Table 1). After *S*. Infantis challenge, the average daily gain (ADG) of the SI group (*p* = 0.002) but not the PS group was significantly decreased. Oral challenge with *S*. Infantis also led to a significant reduction in gain to feed ratio (G:F) (*p* = 0.015), whereas administration of the potential probiotic mixture resisted this effect.

### 3.3. The Effects of Oral Administration of LBB-Mix on Blood Leukocyte Count and Population Distribution

Immediately prior to *S*. Infantis challenge, the total number of peripheral blood leukocytes was not significantly different. Compared with the CN group, an increased peripheral blood leukocyte count was observed at 12 h after *S*. Infantis challenge in the SI group (*p* = 0.025; Figure 2C) but not in the PS group. A continued high percentage of banded neutrophils was observed in the SI group from 6 to 48 h after challenge (*p* = 0.007, *p* = 0.008, *p* < 0.001, and *p* < 0.001, respectively; Figure 2D). Although the percentage of banded neutrophils in the PS group increased at 24 h and 48 h after challenge (*p* = 0.001 and *p* < 0.001, respectively), it was lower than the SI group at 24 h (*p* = 0.049). Compared with the CN group, the SI group exhibited a decrease in the percentage of segmented neutrophils at 24 h after *S*. Infantis challenge (*p* = 0.015; Figure 2E). We also observed a decrease in the proportion of lymphocytes in the SI and PS groups at 48 h (*p* = 0.010 and *p* = 0.036, respectively; Figure 2F).

### 3.4. The LBB-Mix Changes the Composition of Fecal Microbiota during Salmonella Infection

The *Salmonella* detected in the feces of the SI and PS groups persisted from 6 h to 96 h after challenge, only one piglet in the PS group was not detected with *Salmonella* from 72 h. The SI group had a significantly higher *Salmonella* shedding than the PS group at 6 h after infection (*p* = 0.019; Figure 3A). After this, the number of *Salmonella* in the SI group was still higher than that of the PS group, although these differences were not statistically significant.

The number of lactobacilli, *Bacillus* spores, bifidobacteria, coliforms, and enterococci in the feces were monitored using culture-based enumeration (Figure 3B–F). Prior to *S*. Infantis challenge, on day 1, fecal enterococci counts were lower in PS pigs than in CN pigs (*p* = 0.038). Pigs in the PS group had higher lactobacilli and bifidobacteria shedding than that in the SI group on day 4 (*p* = 0.039 and *p* = 0.017, respectively). After *S*. Infantis challenge, on day 8, fecal lactobacilli counts were lower in the SI and PS groups than in the CN group (*p* = 0.018 and *p* = 0.028, respectively). On day 11, fecal enterococci counts were lower in the SI group than in the CN group (*p* = 0.001). Fecal *Bacillus* spores and coliform populations were unaffected by LBB-mix administration with or without *S*. Infantis challenge.

### 3.5. Oral Administration of LBB-Mix Attenuated the Severity of Intestinal Damage and Inflammation Induced by S. Infantis

The *Salmonella*-infected pigs exhibited epithelial cells necrosis and abscission, mucosal hyperemia, submucosa edema, inflammatory cell infiltration, and lymphoid follicle emptying, whereas LBB-mix pretreatment tempered the severity of *Salmonella*-associated pathological injury and inflammation (Figure 4A). In the jejunal tissue, the histologic score between each group was not significantly different (Figure 4B). In the ileum, the histologic score in the SI group was higher than that in the CN and PS groups (*p* < 0.001 and *p* < 0.001, respectively; Figure 4C). Due to weaning stress, pigs in the CN group experienced slight structural damage.

### 3.6. Orally Fed LBB-Mix Increased Goblet Cell Count and Claudin 1 Protein Expression during S. Infantis Infection

During infection, the goblet cells in the PS group showed vacuoles containing a large amount of mucus (Figure 5A). There was no difference between CN and SI pigs with respect to the number of goblet cells in the ileum. However, the number of ileum goblet cells was higher in LBB-mix pretreatment pigs than in CN pigs (*p* = 0.011; Figure 5B). Western blot analysis revealed a reduction in the expression of claudin 1 in the ileum of SI pigs compared with CN pigs (*p* = 0.030; Figure 5C), pretreatment with LBB-mix can save claudin 1 from its decline. However, there were no differences in the expression of occludin among these three groups (Figure 5D).

### 3.7. Orally Fed LBB-Mix Reduces Cell Death in the Intestine after S. Infantis Infection

Compared with the CN group, the DNA damage detected by TUNEL staining in the SI group was upregulated dramatically (*p* = 0.002; Figure 6B). Pretreatment with LBB-mix reduced the degree of DNA damage after *S*. Infantis challenge (*p* = 0.029). The expression level of caspase-1 p10 in the SI group was significantly higher than that in the CN group (*p* = 0.005; Figure 6C), and this increase disappeared by oral administration of LBB-mix. Western blot analysis revealed no difference in the expression of cleaved caspase-4 (Figure 6D) and GSDMD-N (Figure 6E) among the three groups.

## 4. Discussion

The potential probiotic mixture used in the present study was composed of *L. johnsonii* L531, *B. licheniforms* BL1721 and *B. subtilis* BS1715, isolated from clinically healthy weaned piglets and selected based on their ability to survive in the simulated gastrointestinal environments and to reduce the cell death caused by *Salmonella* infection. *Bacillus* is an aerobic bacterium, which uses oxygen in the intestine, providing an oxygen-free environment for colonization of *Lactobacillus*. The present study showed that newly weaned pigs challenged with *S*. Infantis were particularly susceptible, and the sequence of clinical symptoms was generally diarrhea, fever, anorexia, depression, and weight loss. *S*. Infantis caused significant intestinal inflammation in the ileum but not in the jejunum, and oral administration of LBB-mix to pigs could ameliorate ileitis caused by *S*. Infantis infection.

Pathogenic microorganisms such as *Salmonella* have evolved intricate strategies to overcome or manipulate intestinal barrier integrity. This leads to translocation of the pathogen into the lamina propria facilitating further infection of the host and inducing salmonellosis. As an important part of the intestinal barrier, goblet cells secrete mucus, trefoil peptides, and resistin-like molecule-β, which are central to both the defense and repair of the epithelial layer and have significant roles in epithelial homeostasis [25]. Early study has shown that *S*. Typhimurium infection reduces the number of goblet cells in the cecum of the streptomycin-pretreated mice [26]. We did not observe this phenomenon in piglets infected with *S*. Infantis. The LBB-mix administration increased the number of goblet cells during *S*. Infantis infection, thereby increasing the total amount of goblet cell mucin secretion, which not only facilitates the expulsion of *Salmonella* but also maintains the integrity of the mucus layer.

In fact, one of the effects of probiotics exert is to improve the intestinal barrier integrity by strengthening the apical junction complexes of enterocytes and restoring the structures of microfilaments extending into the terminal web. *L. reuteri* I5007 significantly enhanced the protein levels of intestinal epithelial claudin 1, occludin and ZO-1 in newly weaned pigs [27]. Our data indicate that LBB-mix may enhance intestinal epithelial barrier integrity through upregulation of claudin 1 but not occludin. Paradoxically, a previous study showed that *Lactobacillus-* and *Bacillus*-based probiotics obviously increased the protein level of occludin [28], which might be due to different experimental conditions including probiotic strains and animals. Actually, the role of occludin in the assembly and maintenance of TJs is not only related to its expression, but also to different modification events [29].

A complete intestinal barrier reduces *Salmonella* invasion and protects intestinal cells from excessive death. In our study, an increase of TUNEL positive labeling was observed in the ileum of pigs infected with *S*. Infantis only, but not in pigs pretreated with the probiotic mixture. Both apoptotic and pyroptotic cells showed positive TUNEL staining. However, unlike apoptosis, which is generally considered to be non-inflammatory, pyroptosis is a highly inflammatory form of programmed cell death. There is evidence that *Salmonella* infection causes pyroptosis in porcine mesenteric lymph nodes [30]. Since significant ileal inflammation was observed in the SI group, we propose that the infected cells undergo pyroptosis in the ileum, and the probiotic mixture pretreatment can attenuate cell pyroptosis thus prevent damage caused by *Salmonella*.

Caspase-1 is the best-characterized inflammatory caspase and is the central effector protein of the inflammasome. Previous studies indicate that *Salmonella* activates the NLRP3 and NLRC4 inflammasome, resulting in caspase-1 activation and rapid cell death [31,32]. There was evidence that *L. johnsonii* N6.2 suppressed the inflammasome and caspase-1 maturation lowering overall gastro-intestinal inflammation [33]. In our study, increased maturation of caspase-1 was observed in the SI but not in the PS piglets. This result indicated that the canonical inflammasome pathway was activated in the ileum of *Salmonella*-infected pigs, and the probiotic mixture pretreatment can reduce excessive activation of caspase-1 thus prevent intestinal inflammation caused by *Salmonella* infection. 

Various studies have investigated the potential of caspase-4 to initiate innate immune responses to intracellular *Salmonella* infection or *Salmonella*-derived pathogen associated molecular patterns [34,35]. It remains unclear whether porcine caspase-4 represents functional orthologs of human caspase-4, -5 or murine caspase-11. We examined the expression of caspase-4 in the ileum of pigs, but there was no difference in cleaved caspase-4 among the three groups, whether caspase-4 can play a role in the infection of *Salmonella* in pigs remains to be further explored.

GSDMD is an executioner of pyroptosis owing to its ability to be cleaved by inflammatory caspases and its ability to form membrane pores. Either caspase-1 or caspase-4 independently cleaves GSDMD, from which the released amino-terminal fragment associates with the cell membrane and oligomerizes to form the pyroptotic pore. The cell then swells, resulting in membrane rupture that is known as pyroptosis. Triggering pyroptosis is a property shared by multiple *S*. *enterica* serovars. Unexpectedly, in our study, we did not observe the obvious cleavage GSDMD in the ileum of *Salmonella*-infected pigs, and the Western blot analysis revealed no difference in the expression of GSDMD-N among the three groups. It is probably because pyroptosis is a relatively early event and our sampling time is later. *Salmonella* can evade the host immune system by down-regulating the expression of PrgJ or flagellin. Previous studies in mice have suggested that *S*. Typhimurium efficiently evade inflammasomes during the systemic phase of infection [36,37]. Another possible explanation is that other Gasdermin family members induced pyroptosis in the ileum of pigs infected with *Salmonella* [38].

## 5. Conclusions

In conclusion, our data suggest that a select mixture of *Lactobacillus* and *Bacillus* is effective in preventing *Salmonella* infection in newly weaned pigs by maintaining the intestinal mucosal barrier integrity and reducing intestinal cell death, although it does not prevent diarrhea. Our findings provide a new preventative strategy that can ameliorate *S*. Infantis-induced ileal inflammation in pigs and thus help to reduce the use of antibiotics. However, the differences in the effects between probiotic mixture and single strains were not compared in the current study. More research is still needed to obtain symbiotic or synergistic combinations to maximize the positive benefits of these probiotic combinations.

## Figures and Tables

**Figure 1 microorganisms-07-00135-f001:**
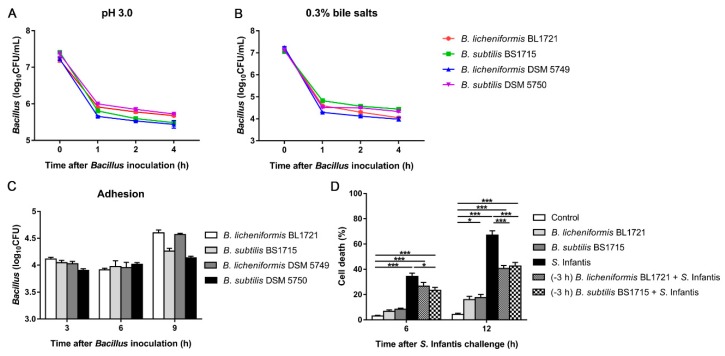
In vitro assessment of *Bacillus* for use as livestock probiotics. Survival of *Bacillus* in pH 3.0 (**A**) and 0.3% porcine bile salts (**B**). (**C**) The ability of *Bacillus* to adhere to epithelial cells. (**D**) Trypan blue staining was used to investigate cell death, the results are presented as the ratio of the number of dead cells to the total number of cells. All analyses were performed in triplicate. * *p* < 0.05; ** *p* < 0.01; *** *p* < 0.001.

**Figure 2 microorganisms-07-00135-f002:**
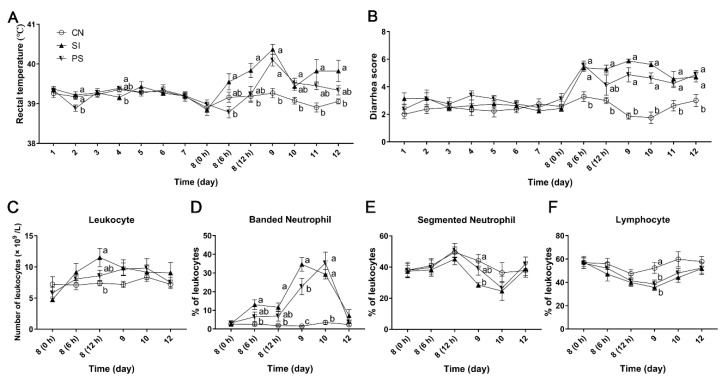
Effects of *Lactobacillus johnsonii* L531, *Bacillus licheniformis* BL1721 and *Bacillus subtilis* BS1715 (LBB-mix) on clinical signs and blood cell differential counts of *Salmonella* Infantis infection. (**A**) Rectal temperature and (**B**) diarrhea scores in pigs (*n* = 8 per group). (**C**) The total number of peripheral blood leukocytes. The percentage of banded neutrophils (**D**), segmented neutrophils (**E**), and lymphocytes (**F**) on blood leukocytes. Control group with sterile physiological saline (CN); oral sterile physiological saline from day 1 to day 7 followed by *S*. Infantis challenge on day 8 (SI); Pretreated with the potential probiotic mixture for day 1 to day 7 and followed by *S*. Infantis challenge on day 8 (PS). Mean values at the same time point without a common superscript (^a, b, c^) differ significantly (*p* < 0.05).

**Figure 3 microorganisms-07-00135-f003:**
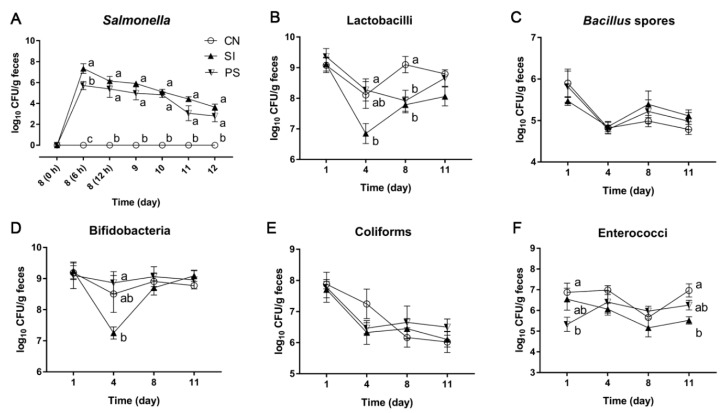
Effects of LBB-mix on the composition of fecal microbiota during *S*. Infantis infection. (**A**) *Salmonella*, (**B**) lactobacilli, (**C**) *Bacillus* spores, (**D**) bifidobacteria, (**E**) coliforms and (**F**) enterococci. Fresh fecal samples from all the piglets were collected on days 1, 4, 8 and 11 after weaning. Results are presented as log_10_ CFU/g feces, and all counts were performed in triplicate. Data are presented as the mean ± SEM (*n* = 8 per group). Within the same time, mean values with different superscript letters (^a, b^) are significantly different (*p* < 0.05).

**Figure 4 microorganisms-07-00135-f004:**
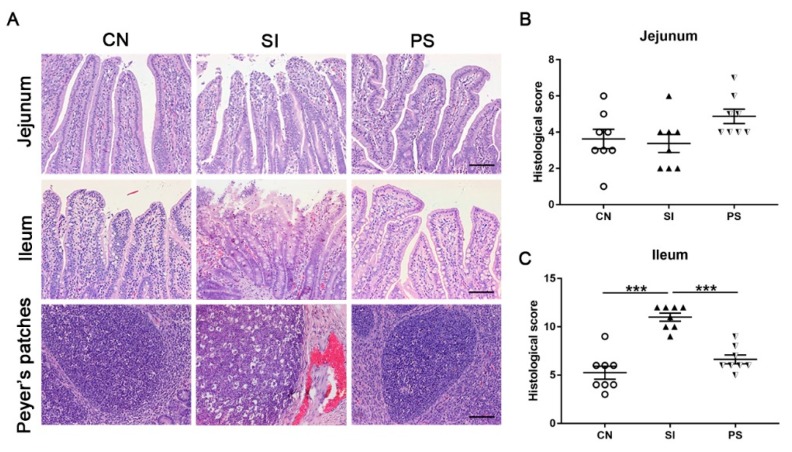
Effects of LBB-mix on the small intestinal inflammation. (**A**) Representative photomicrographs of hematoxylin and eosin–stained jejunum and ileum sections. Scale bars, 100 µm. (**B**,**C**) Jejunal and ileal histologic scores. Data are presented as the mean ± SEM (*n* = 8 per group). * *p* < 0.05; ** *p* < 0.01; *** *p* < 0.001.

**Figure 5 microorganisms-07-00135-f005:**
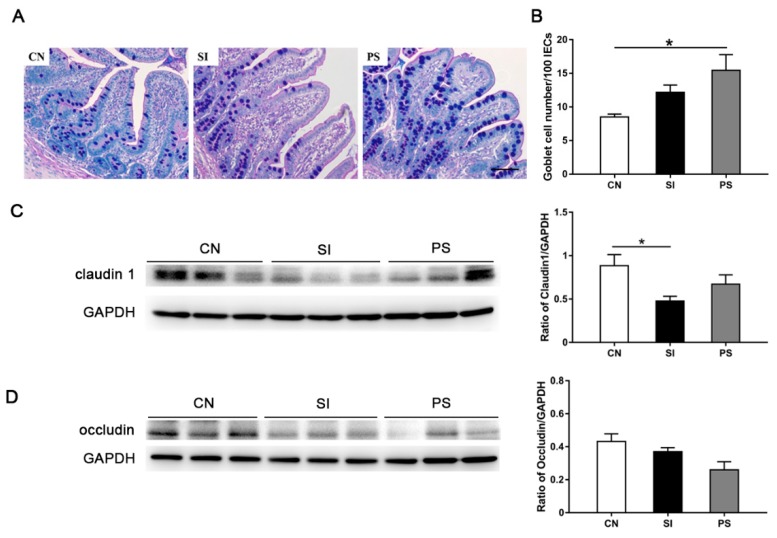
Effects of LBB-mix on the intestinal mucosal barrier in the ileum. (**A**) Goblet cells in ileal tissues determined by alcian blue/periodic acid–Schiff (AB/PAS) staining, scale bar, 100 µm. (**B**) Numbers of goblet cells in the ileum. Representative Western blot results for claudin 1 (**C**) and occludin (**D**) in ileal tissues. Each band represents a single pig. Data are presented as the mean ± SEM (*n* = 8 per group). * *p* < 0.05; ** *p* < 0.01; *** *p* < 0.001.

**Figure 6 microorganisms-07-00135-f006:**
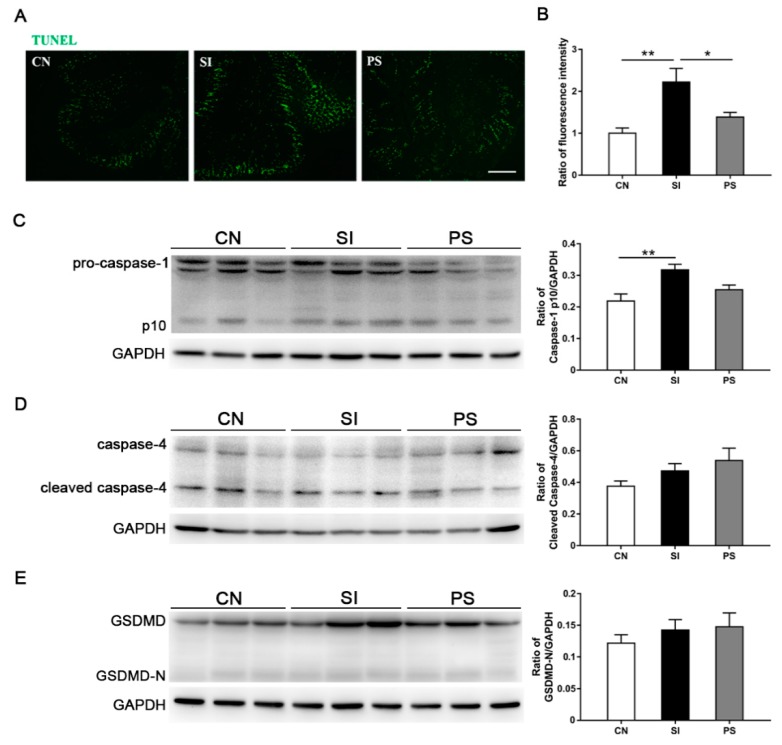
Cell death analysis of the ileum from *S*. Infantis-infected pigs. (**A**) Terminal deoxynucleotidyl transferase dUTP nick end labeling (TUNEL) analysis in the ileum. Scale bar, 100 μm. (**B**) Quantification of TUNEL fluorescent labeling. Representative Western blot results for caspase-1 (**C**), caspase-4 (**D**), and gasdermin-D (GSDMD) (**E**) in ileal tissues. Each band represents a single pig. Data are presented as the mean ± SEM (*n* = 8 per group). **p* < 0.05; ***p* < 0.01; ****p* < 0.001.

**Table 1 microorganisms-07-00135-t001:** Effects of oral administration of LBB-mix on the growth performance of weaned piglets before and after *S*. Infantis challenge.

Item ^1^	Treatments ^2^			SEM	*p* Value
	CN	SI	PS		
**Pre-challenge (days 1 to 7)**					
BW, kg (day 1)	4.656	4.894	5.263	0.177	0.387
ADG, g/d	136.607	143.75	149.107	10.588	0.898
G:F, g/g	0.615	0.701	0.612	0.048	0.707
**Post-challenge (days 8 to 13)**					
BW, kg (day 13)	6.3	5.85	6.443	0.211	0.379
ADG, g/d	137.5 ^a^	−10 ^b^	55 ^ab^	19.476	0.003
G:F, g/g	0.495 ^a^	−0.049 ^b^	0.21 ^ab^	0.083	0.019

^1^ Item: BW, body weight; ADG, average daily gain; G: F, gain to feed ratio; ^a, b^ within a row, means without a common lowercase superscript differ (*p* < 0.05); Turkey’s test; ^2^ Treatments: CN, control group with sterile physiological saline; SI, oral sterile physiological saline from day 1 to day 7 followed by *S*. Infantis challenge; PS, pretreated with potential probiotic mixture for day 1 to day 7 and followed by *S*. Infantis challenge; *n* = 8 per group.

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
