# Peer review of "Oral Administration of a Select Mixture of Lactobacillus and Bacillus Alleviates Inflammation and Maintains Mucosal Barrier Integrity in the Ileum of Pigs Challenged with Salmonella Infantis"

_microorganisms, 2019, doi:10.3390/microorganisms7050135_

Round 1

Reviewer 1 Report

An interesting and well presented study.

Only a few minor comments

Line 50 lamina to be changed to lamina propria

Figure 4a Differences in histology scores are not clearly reflected in the chosen histology image.

Figures 5 and 6. What do the 3 lanes in each group represent? Individual samples? Need to indicate in legend what these are.

Line 326 'Bacillus is aerobes'  scientific English could be improved.

Line 340 'massive mucus' again this could be improved

A non breaking space should be used in bacterial names to prevent the name dividing over 2 lines.

Author Response

Point 1: Line 50 lamina to be changed to lamina propria

Response 1: Thank you for providing us the valuable comments. In the revised manuscript, the “lamina” has been changed to “lamina propria”. (line 55)

Point 2: Figure 4a Differences in histology scores are not clearly reflected in the chosen histology image.

Response 2: We agree with the comments. In the revision, Figure 4a has been replaced by a more representative picture.

Point 3: Figures 5 and 6. What do the 3 lanes in each group represent? Individual samples? Need to indicate in legend what these are.

Response 3: Thank you for providing us the valuable comments. The 3 lanes in each group represent 3 individual samples. In the revised manuscript, we indicated what the lanes are.

Figure 5: “Representative Western blot results for Claudin 1 (C) and Occludin (D) in ileal tissues. Each band represents a single pig.” (lines 318-319)

Figure 6: “Representative Western blot results for Caspase-1 (C), Caspase-4 (D), and GSDMD (E) in ileal tissues. Each band represents a single pig.” (line 330-331)

Point 4: Line 326 'Bacillus is aerobes' scientific English could be improved.

Response 4: We agree with the reviewers' comments. In the revised manuscript, It has been changed to “Bacillus is an aerobic bacterium”. (line 337)

Point 5: Line 340 'massive mucus' again this could be improved.

Response 5: We agree with the reviewers' comments. In the revised manuscript, The statement has been re-written as “…thereby increasing the total amount of goblet cell mucin secretion”. (lines 352-353)

Point 6: A non breaking space should be used in bacterial names to prevent the name dividing over 2 lines.

Response 6: Thank you for the comments. In the revision, we have carefully checked the sentences and used non breaking space in bacterial names to prevent the name dividing over 2 lines.

Reviewer 2 Report

Liu et al. wrote an original article concerning very interesting and up-to date problem. Finding new strategies preventing salmonellosis in meet production can reduce the use of antibiotics in global scale. The article is well written, and the methodology was correctly matched, however I have some suggestions, listed below:

line 36-38: The authors should  give additional information: What are the actual recommendations for salmonellosis treatment? Which countries allow using antibiotics in pork production, and which not? What are other options for controlling Salmonella infections (including bacteriophages)?

line 126-138: Why the experiment was conducted for 13 days? 

line 208: large amounts – authors should indicate the range of number of cells. 

Figures 2-3 should be bigger, as it is hard to compare the lines.

line 390-397: In conclusions the Authors should also mention that a select mixture of probiotics did not prevent the piglets from diarrhoea. 

Author Response

Point 1: line 36-38: The authors should give additional information: What are the actual recommendations for salmonellosis treatment? Which countries allow using antibiotics in pork production, and which not? What are other options for controlling Salmonella infections (including bacteriophages)?

Response 1: Thank you for providing us the valuable comments. In the revised manuscript, we have added “Conventional salmonellosis control strategies have relied on antibiotics. However, due to the emergence of drug-resistant strains, many countries have already restricted antibiotic use in animal agriculture. In 2006, the European Union banned the use of antimicrobial growth promoters in animal food and water [5]. In recent years, different intervention strategies such as vaccination, antimicrobial peptides, nutritional supplements, bacteriophages and probiotics have been used to control Salmonella infection as antibiotic alternatives [6].” (lines 36-41)

Point 2: line 126-138: Why the experiment was conducted for 13 days?

Response 2: One aim of this study was to determine the ability of Lactobacillus and Bacillus mixture to prevent S. Infantis infection. Thus, we started to oral administration of LBB-mix 7 days before S. Infantis challenge. Another aim of this study was to investigate the effect of S. Infantis infection on intestinal cell pyroptosis in pigs. Previous study has shown the cell death in mesocolic lymph nodes was peaked at 1 dpi and decreased rapidly [30]. In addition, some piglets in the SI group showed negative Salmonella culture in feces from day 11 in this experiment. Therefore, we chose to slaughter and sample on day 13. In the future study, we will try to sample at multiple time points.

Point 3: line 208: large amounts – authors should indicate the range of number of cells.

Response 3: We agree with the reviewers' comments. In the revised manuscript, the statement has been replaced by “The number of adhere B. licheniformis BL1721 was 1.31 × 104 ± 9.50 × 102 CFU (the mean ± SEM) at 3 h after inoculation, and increased to 4.06 × 104 ± 5.46 × 103 CFU at 9 h. The number of B. subtilis BS1715 was 1.13 × 104 ± 1.10 × 103 CFU at 3 h after inoculation, and increased to 1.86 × 104 ± 2.28 × 103 CFU at 9 h (Figure 1C). Both isolates and commercial strains have the similar adhesion ability to IPEC-J2 cells.” (lines 214-218)

Point 4: Figures 2-3 should be bigger, as it is hard to compare the lines.

Response 4: We agree with the reviewers' comments. In the revised manuscript, Figure 2-3 have been enlarged.

Point 5: line 390-397: In conclusions the Authors should also mention that a select mixture of probiotics did not prevent the piglets from diarrhoea.

Response 5: Thank you for the comments. In conclusions, we have added “…although it does not prevent diarrhea.” (line 404)

Reviewer 3 Report

The authors described the effects of the oral administration of a select mixture of Lactobacillus and Bacillus on the inflammation and the mucosal barrier integrity in the ileum of pigs challenged with Salmonella Infantis. This manuscript is overall well-written, clearly presented and easy-to-read, also providing novel, useful information for the scientific community. I have only some minor, formal suggestions for the authors, in oder to make the paper suitable for the publication on the journal.

1) Please, add the article before the abbreviations when you use them at the beginning of a sentence.

2) Materials and Methods:

- Line 84: please, replace "was" with "were".

- Lines 160-161: please, specify the exact location of the gut sampling (it is important to assess if the sampling has been standardized). I also have a specific question: why have you also sampled the jejunum for the histopathological examination if all the other analyses have been performed on the ileum only? Perhaps you can remove it or (as I suggest) highlight and discuss the differences you observed between these two segments in relation to the inflammation (that was attenuated only in the ileum after LBB-mix administration) in the Discussion section.

3) Discussion:

- Lines 339-342: please, re-write the sentence (it can be split into two different sentences).

- Line 383: please, remove the full stop after "pigs".

- Lines 385-388: please, re-write the sentence (it can be split into two different sentences).

Author Response

Point 1: Please, add the article before the abbreviations when you use them at the beginning of a sentence.

Response 1: Thank you for providing us the valuable comments. In the revision, we have carefully checked the sentence that were missing the article and added them to lines 104, 111, 273 and 351.

Point 2: Materials and Methods:

- Line 84: please, replace "was" with "were".

- Lines 160-161: please, specify the exact location of the gut sampling (it is important to assess if the sampling has been standardized). I also have a specific question: why have you also sampled the jejunum for the histopathological examination if all the other analyses have been performed on the ileum only? Perhaps you can remove it or (as I suggest) highlight and discuss the differences you observed between these two segments in relation to the inflammation (that was attenuated only in the ileum after LBB-mix administration) in the Discussion section.

Response 2: We appreciate the reviewer’s concern and comments.

- Line 90: It has been changed to “were”.

- Lines 165-166: We added the following statement according to the reviewer’s comment. “The proximal, middle, and distal segments (approximately to 10 × 15 × 3 mm) of the jejunum and ileum were rinsed with sterile physiological saline and then fixed in 4% paraformaldehyde.”

- Lines 341-343: About the choice of samples, histopathological examination showed that oral challenge with S. Infantis can not cause significant damage and inflammation in the jejunum. Therefore, we did not detect the relevant indicators of the jejunum in the following experiments. In the discussion section, we added the following statement “S. Infantis caused significant intestinal inflammation in the ileum but not in the jejunum, and oral administration of LBB-mix to pigs could ameliorate ileitis caused by S. Infantis infection.”

Point 3: Discussion:

- Lines 339-342: please, re-write the sentence (it can be split into two different sentences).

- Line 383: please, remove the full stop after "pigs".

- Lines 385-388: please, re-write the sentence (it can be split into two different sentences).

Response 3: Thank you for the comments. In the revised manuscript, the statements have been re-written.

- Lines 351-354: “The LBB-mix administration increased the number of goblet cells during S. Infantis infection, thereby increasing the total amount of goblet cell mucin secretion, which not only facilitates the expulsion of Salmonella but also maintains the integrity of the mucus layer.”

- Line 394: The full stop after “pigs” has been removed.

- Lines 396-399: “Salmonella can evade the host immune system by down-regulating the expression of PrgJ or flagellin. Previous studies in mice have suggested that S. Typhimurium efficiently evade inflammasomes during the systemic phase of infection [36,37].”
